# Electrical Stimulation for Preventing Skin Injuries in Denervated Gluteal Muscles—Promising Perspectives from a Case Series and Narrative Review

**DOI:** 10.3390/diagnostics13020219

**Published:** 2023-01-06

**Authors:** Marie Alberty, Winfried Mayr, Ines Bersch

**Affiliations:** 1International FES Centre^®^, Swiss Paraplegic Centre, Guido A. Zäch Strasse 1, CH-6207 Nottwil, Switzerland; 2Medical University of Vienna, Center for Medical Physics and Biomedical Engineering, Währinger Gürtel 18-20, AT-1090 Vienna, Austria

**Keywords:** spinal cord injury, lower motor neuron lesion, electrical stimulation, magnetic resonance imaging, long-term denervation, pressure ulcers, prevention

## Abstract

Spinal cord injury (SCI) where the lower motor neuron is compromised leads to atrophy and degenerative changes in the respective muscle. This type of lesion becomes especially critical when the gluteal muscles and/or the hamstrings are affected as they usually offer a cushioning effect to protect from skin injuries. Previous research conducted over the past 30 years has made advancements in the development of parameters for the optimal application of long pulse stimulation with the aim to restore muscle structure and trophic aspects in people with chronic SCI (<20 years post-injury). This work provides an overview of previous achievements in the field through a narrative literature review before presenting preliminary results in the form of a case series from an ongoing study investigating the acute effects of six months of long pulse stimulation on the tissue composition of the gluteal muscles in five people with chronic SCI (>20 years post-injury). Participants underwent a 33-min home-based long pulse stimulation program five times a week, and their muscle and adipose tissue thicknesses were assessed at baseline, after three and six months, respectively, using magnetic resonance imaging. The results show that the largest increase in muscle thickness occurred at the level of the height of the acetabulum (+44.37%; χ^2^(2) = 0.5; *p* = 0.779), whereas the most important decrease in adipose tissue occurred at the level of the sacroiliac joint (SIJ) reference (−11.43%; χ^2^(2) = 1.6; *p* = 0.449) within only six months of regular stimulation despite the preceding long denervation period. The underlying mechanism and physiology of muscular resuscitation from myofibrillar debris as presented in chronic denervation to functional contractile entities remain to be investigated further.

## 1. Introduction

Damage to the lower motor neuron (LMN) after spinal cord injury (SCI) has detrimental consequences in the years that follow the injury. Within months, muscles begin to atrophy and with the absence of neural input, the denervated muscle starts to degenerate. The degeneration process includes several stages occurring over the course of years after injury according to Carlson [1] starting with a reduction in muscle mass by about 40% within two years of denervation [2], accompanied by further atrophy and disorganization of the muscular ultrastructure at the T-tubules after four months and the contractile entities or sarcomeres. Finally, these changes lead to a degeneration of myofibers and lastly to a replacement of muscle by adipose and connective tissue that is accompanied by a decrease in tissue vascularization. Moreover, the degeneration process is accompanied with a severe loss of muscle volume in the paralyzed muscles. Notably in the chronic phase after SCI a consequence of muscle inactivity is a disarrangement of intermyofibrillar mitochondria providing the energy for muscle force production [3]. Considering the multitude of risk factors present in people with SCI, and the fact that a majority is wheelchair-bound, an LMN lesion in the gluteal region puts them at a major risk of developing tissue damage [4]. Causes of the development of pressure injuries in the SCI population are multifactorial including time since injury, aging, gender, weight (over- and underweight), comorbidities, smoking, social context, and the nutritional status [5]. These defined risk factors are in accordance with the worldwide non-SCI population developing a pressure injury [6].

In Switzerland, pressure ulcers present a major complication after SCI with an average of 2.2 pressure ulcers per person year and with the sacrum, ischium and trochanter being the most severely affected areas [4]. Not only does this loss in muscle mass affect the metabolism in a negative way, but it also constitutes a major problem in regard to skin protective mechanisms. Furthermore, the absence of muscle contraction results in a lack of traction on the bone, thereby creating a risk for experiencing bone fractures due to a decrease in bone mineral density (BMD) [7]. Interventions involving the application of electrical stimulation (ES) in order to counteract these complications have been thoroughly investigated and it is evident that ES offers an auspicious perspective to partially or fully recover muscle and bone structure and further provide improvement in cardiovascular health [8,9]. 

ES might be effective as research has proposed that some muscle fibers persist even three to six years after SCI [10]. Those muscles affected by an LMN lesion cannot be activated by ES in the same way as muscles with an upper motor neuron (UMN) lesion since the ultrastructure is not maintained. In addition, LMN lesions provoke changes in muscular and adjacent soft tissue, involving fat substitution and fibrotic transformations within months after denervation [5]. Therefore, special current forms need to be utilized when applying ES and as research has proven in recent decades, stimulation with long pulse durations seems to be the most effective way to recruit muscle tissue directly, after motor nerve supply got lost. In order to detect muscle denervation, Kern & Carraro [11] recommend applying neuromuscular stimulation with a pulse width ranging from 0.3–1 milliseconds [ms] and a frequency of 40 Hertz [Hz] which will elicit a fused/tetanic muscle contraction in case of an intact LMN. The amplitude is chosen according to the size of the muscle but is generally safe to apply up to 100 milliamperes [mA].

ES of denervated muscles with long pulses to alter connective and adipose tissue into contractile muscle tissue has been investigated in persons with chronic spinal cord injury [11,12,13,14,15]. As it has been shown through the EU-project RISE initiated at the beginning of the 2000s, two years of home-based ES resulted in a significant gain of muscle thickness and reversed the muscle loss in relation to other soft tissue components [11,14,16,17,18]. Especially if started early after SCI, the number and size of myofibers that can be saved through ES, are higher [19]. In particular, a pronounced reduction of adipose tissue in and around the lower limb muscles has been observed [15,20]. Positive complementary effects of ES for denervated muscles include the increase in epidermal thickness and morphology to near normal values [21].

Nonetheless, all the positive changes observed after regular ES therapy along two years or more, leave the question about the necessary application time to guarantee a successful reversing transformation from connective and fatty tissue into contractile muscle tissue in people with damage to the LMN unanswered. Likewise, the exact process of this restoration, how it occurs and under which conditions it develops best, remains unclear. In other words, is there an immediate visible increase in number and size of muscle fibers first or does connective and adipose tissue become degraded first or do both processes occur simultaneously? Moreover, how long is the delay between start of intervention and onset of substantial tissue changes? Kern and Carraro [22] have visualized and emphasized on the importance of life-long stimulation in people with lower motor neuron lesion in order to maintain proper blood circulation and to produce a so-called "cushioning effect" in people with SCI to prevent them from skin injuries. Addressing the above-mentioned questions, the aim of the present study is to investigate the effect on tissue composition and its time-course through the first six months after stimulation onset, when ES treatment is started years after denervation of gluteal muscles in chronic-state demonstrating degeneration changes. In this work, we would like to present preliminary results in form of a case series including the first five study participants.

## 2. Materials and Methods

### 2.1. Narrative Literature Search

In order to evaluate the effects of electrical stimulation (ES) in denervated muscles in people with SCI and outline previous work in this field, a narrative literature search in the Pubmed database was performed, combining the search terms “denervation”, “spinal cord injuries” and “electrical stimulation” or “long-pulse stimulation” in various ways adding a subsequent filtering of the results restricting it to human studies conducted after the year 2000.

### 2.2. Case Series

#### 2.2.1. Participant Eligibility

The protocol of the interventional study was accepted and approved by the ethical committee of Northwest- and Central Switzerland (EKNZ) and is conducted in compliance with the Declaration of Helsinki. It has been registered as a clinical trial under the identifier number NCT02080039. Study participants have been recruited at the Swiss Paraplegic Centre (Nottwil, Switzerland) as of March 2020. Inclusion criteria for this study are outpatients with a chronic SCI (>1 year after injury), aged 18–70 years, a neurological level of injury between T10 and L5 classified AIS A-D as evaluated by the International Standards for Neurological Classification of SCI (ISNCSCI) and presenting with a denervation of the gluteal musculature. The screening for gluteal muscle denervation has been conducted through bilateral application of neuromuscular ES on the m. gluteus maximus using oval 7.5 cm × 13 cm self-adhesive PALS^®^ surface electrodes (Axelgaard Manufacturing Co., Ltd, Fallbrook, CA, USA). Using a current frequency of 50 Hz, a pulse duration of 300 μs and an amplitude of 100 mA, two experienced and trained therapists, being part of the study staff, stimulated the gluteal muscles. In case where no contraction could be elicited in response to these high stimulation currents, muscles were identified as denervated. Exclusion criteria compromised people presenting arteriosclerosis, an acute pressure injury, skin irritation, infection, or recent plastic surgery (<3 months) on the area of stimulation of the gluteal muscles. All participants screened positively signed an informed consent form before study participation.

#### 2.2.2. Intervention

For the home-based training, the participant was lying in prone position while ES was administered with the Stimulette den2x (Schuhfried GmbH, Vienna, Austria). Sponge pockets with the corresponding electrodes sized 10 cm × 13 cm were placed on the buttocks to generate a horizontally oriented electrical field, conducted through one channel, to cover the entire gluteal area. An elastic fabric band was given to the participants to render home-usage and electrode fixation easier (Figure 1).

ES was conducted five times a week for five consecutive days followed by two days break over a period of six months. The stimulation session consisted of a warm-up phase of three minutes and a training phase of 30 min. Hence, including pre- and post-preparation time, 45 min of time expenditure can be calculated. The parameters for the stimulation of denervated muscles have been proven for efficacy and were used in previous work with the long pulse width being specific to elicit contractions in denervated fibers [22]. In contrast to previous projects, where the pulse width was adapted according to the excitability of the muscle, the same parameters were maintained throughout the study. The stimulation was divided into a warm-up and training phase, with the warm-up phase serving to reduce the discharge rate of motor units, increase the excitability of the muscle fibers and adapt the skin to the following intensive training phase stimulation [23,24]. The stimulation parameters chosen for the warm-up were 11 s bursts with a frequency of 0.86 Hz, biphasic ramp-shaped pulses with impulse durations of 150 ms (75 ms per phase) and inter-pulse pauses of 1 s followed by 11 s breaks. The training phase was conducted with a biphasic symmetrical rectangular current and a frequency of 20 Hz, 40 ms impulse durations (20 ms per phase) and 10 ms inter-pulse pauses administered with 2 s bursts and 2 s breaks (Figure 2). Participants were instructed to stimulate at 90 mA and 120 mA during the warm-up and training phase, respectively, unaltered throughout the study.

#### 2.2.3. Outcomes Measures

Changes in muscle as well as adipose tissue thickness were assessed over the stimulation period of six months using standardized magnetic resonance imaging (MRI) measurements with a 3 Tesla device (3T-MRI Archeva, Philips, Horgen, Switzerland) of the gluteal area on the projected line of the posterior superior iliac spine (PSIS) neighboring the sacroiliac joint (SIJ), the acetabulum and the coccyx. During the MRI measurement, the participant was in a prone-lying position. Measurements were performed at baseline, after three months and after six months following stimulation onset as shown in Figure 3. The same radiology expert determined the quantitative values of tissue thicknesses by measuring the adipose and muscle tissue thicknesses in millimeters within the T2 sequence using the Philips Intellispace Portal software (Philips N.V., The Netherlands). Moreover, weekly phone calls were performed to provide support and motivation as well as warrant a safe and compliant implementation according to the study protocol. Throughout their study participation, people completed a stimulation diary that comprised information about the date, duration, amplitude and a comment section.

#### 2.2.4. Statistical Analysis

Statistical analyses were performed using SPSS IBM statistics version 29.0 and reported as median ± interquartile ranges. After checking for normal distribution of the data using the Shapiro-Wilk test, the Friedman test for repeated measures was performed to analyze changes in muscle and adipose tissue thickness among all participants between the three measurement timepoints for each anatomical area described above. The statistical significance level was set at *p* < 0.05.

## 3. Results

The detailed search strategy is illustrated in Figure 4 and retrieved 16 relevant articles that are summarized in the table in the Appendix A.

For the preliminary analysis of this study, five study participants have been included. Patient characteristics are illustrated in Table 1.

As the test for normal distribution was not passed, the non-parametric Friedman test for repeated measures was applied to the dataset. Our results, depicted in Figure 5 as median and interquartile ranges, show an increase in gluteal muscle tissue thickness at the level of the acetabulum (+0.10 mm and +2.30 mm between baseline-3 months and 3–6 months respectively; total change: +12.47%, χ^2^(2) = 1.37; *p* = 0.504) and coccyx (+3.50 mm and −0.35 mm between baseline-3 months and 3–6 months respectively; total change: +44.37%; χ^2^(2) = 0.5; *p* = 0.779), but not at the level of the SIJ. In fact, a slight decline in muscle tissue thickness could be measured at the projected line of the PSIS (−0.55 mm and −0.55 mm as measured from baseline-3 months and 3–6 months respectively; total change: −12.79%; χ^2^(2) = 3.5; *p* = 0.174).

Regarding adipose and connective tissue thicknesses, there was a reduction at the height of the SIJ (−5.55 mm and −4.90 mm between baseline-3 months and 3–6 months respectively; total change: −11.43%; χ^2^(2) = 1.6; *p* = 0.449) whereas adipose tissue thickness increased at the acetabulum level (+2.35 mm and +0.90 mm between baseline-3 months and 3–6 months respectively; total change: +12.31%; χ^2^(2) = 0.105; *p* = 0.949). At the level of the coccyx, no changes could be observed over the entire study period (+4.30 mm and −4.30 mm between baseline-3 months and 3–6 months respectively; total changes: ±0%; χ^2^(2) = 0.4; *p* = 0.819). Hence, the changes in adipose tissue thickness align with the changes in muscle thickness as all changes were not statistically significant.

## 4. Discussion

In the present work, we have shown that even throughout the first six months of ES in very chronically denervated muscle of people with LMN lesions, there are transformations taking place at the muscular and adipose tissue levels. Through these preliminary data, we provide evidence that might be indicative of the possibility to stimulate very chronically denervated (>20 years) and degenerated muscles and consequently elicit changes in tissue composition. It is undebatable that an early administration of ES in people in whom denervation has been diagnosed is most effective in preventing muscle denervation atrophy or recovering muscle tissue (almost) completely.

Overall, our results show that the tissue thickness measured through MRI at the three defined levels is constituted by a larger extent of adipose tissue compared to muscle tissue. Muscle tissue thickness in the gluteal area increased at the projected heights of the acetabulum and the coccyx of the prone-lying participant who has stimulated using long pulse stimulation over the course of six months. Simultaneously, connective, and adipose tissue thickness decreased at the level of the SIJ. At the coccyx level, muscle tissue thickness increased while the thickness of adipose and connective tissue remained roughly the same between baseline and six months into stimulation. We see that changes in tissue composition do not seem to occur in a specific order but rather simultaneously as there were both changes in muscle as well as adipose tissue thicknesses. Hence, the preliminary results could not provide statistical significance but important clinical evidence for the treatment with ES. It is worth mentioning that during the weekly phone calls, none of the participants reported difficulties with the administration of the stimulation or the fixation of the electrodes. Some reported a positive effect on hip mobility as the prone-lying positioning imposes a stretch on the hip flexors.

Applying long pulse stimulation (40 ms) at higher frequencies (20 Hz) did not lead to a visible contraction from the beginning. As the chronaxie increases due to denervation and increased impedance caused by accumulation of adipose tissue, the muscle can initially only be activated through twitch stimulation at low frequencies (2 Hz) with long pulse durations (200 ms). In four of the five described cases, it was not possible to get any contractions from the beginning, even with the warm-up single twitches. In all participants, muscle contraction could be elicited at some point within the six months of the stimulation period, which might be an indication of functional regeneration.

Considering the literature, most of the research on the morphological changes in denervated muscles through the administration of direct muscle stimulation has been carried out by the research group of Helmut Kern and Ugo Carraro, who also played a leading role in the RISE project. All studies conclude that long pulse stimulation increases CSA, improves torque, and reduces fat infiltration in denervated muscle tissue [2,13,15,17,20,25,26]. This is accompanied by improvements in muscle density and regeneration of muscle fibers. Furthermore, two studies investigated the thickness of the epidermis in the person suffering from permanent denervation of leg muscles as a result of a complete cauda equina lesion. Albertin and colleagues highlighted an increase in skin epidermis thickness from to 47.6 μm ± 8.8 μm before electrical stimulation compared to 60.8 μm ± 12.7 μm after 2 years of every day electrical stimulation [21,27]. All these results taken together support the recommendation to start long pulse stimulation in a timely manner after injury where the LMN’s integrity is compromised. This remains true, even if, based on the current results, there is an indication that contractility of the musculature can still be achieved even after 20 years of denervation atrophy.

The stimulation protocol used in the current study differs from the studies listed in the narrative review in that a warm-up phase of 3 min with a single twitch stimulation was consistently applied, followed by the training phase with a tetanic contraction. This was performed regardless of an already visible contraction. The papers reviewed performed single twitch stimulation until a noticeable muscle response was seen over the treatment session before applying higher frequency stimulation (20 Hz), ensuring a tetanic contraction. A recently published study by Bersch and colleagues [28] demonstrated in the upper limb that even with chronic damage to the LMN, a contraction can be achieved within 12 weeks after stimulation with the protocol applied in the current study. This approach could ultimately be confirmed in the observations of the present study.

To our knowledge, this is the first time since the EU-RISE project that long-term chronic denervated muscles regaining muscle contractions could be observed.

### 4.1. Limitations

It should be noted that these are preliminary data from a study including only five participants and that the effects of ES were heterogeneous regarding the individual changes in tissue thicknesses. Furthermore, measurements were performed at the same three levels for each participant despite the chronicity of SCI. As many people with SCI present comorbidities, such as scoliotic deformations or fractures, these bony landmarks might not relate to the same anatomical section in the investigated muscle.

Furthermore, a method investigating CSA instead of thickness might be more sensitive to changes, especially in this early stimulation as well as very chronic denervation phase, where it is evident that muscle takes a long time to regenerate.

### 4.2. Future Perspectives

Future imaging techniques like colour three-dimensional computer tomography imaging proposed by Gargiulo and coworkers [29,30,31,32,33] or fiber tracking, and the determination of the fractional anisotropy factor should be considered. These methods may offer more detailed quantification of the amount of connective and adipose tissue surrounding the muscle but also of the intramuscular fat infiltration.

In contrast to the RISE project protocol, our parameters for ES were kept constant in this study. After completion and publishing of the results of the whole sample, it should be investigated what the minimal dose of ES to maintain muscle mass and microstructure in acute SCI with damage to the LMN is and how parameters of ES should be adapted over time. Is it most efficient to introduce ES using muscle twitches only and continuing building it up with tetanic contractions or should tetanic contractions be implemented from the beginning? Electrophysiological measurement might be implemented in this investigation. It is well described how denervation progresses over time and how it affects the different muscle tissue components. However, when it comes to the progressive regeneration of muscle tissue, we only have data after one year of stimulation. Future research should focus on more acute effects of ES on denervated muscle in people with SCI, once more because the untrained muscle will be more prone to muscle fatigue and parameters will need to be more closely monitored in this critical phase for optimal build-up with the least possible damage due to over-training. Additionally, a follow-up of long-term ES users may provide important information if the incidence of pressure injuries can be reduced in comparison to non-ES users. Henceforth stimulation of denervated gluteal muscle might become a standard of care treatment in preventing pressure injuries.

## 5. Conclusions

There is a strong body of research that has investigated the effect of ES on denervated muscle. It is recommended to initiate treatment with a special scope on prevention as early as possible after denervation, independent of the etiology. As we are able to elicit muscle contractions in people with chronic SCI (>20 years) with severely denervated and degenerated muscles, the underlying mechanisms should be investigated more in depth at the cellular and metabolic level. This work should also serve as an encouragement to make use of ES as therapeutic or preventive strategy no matter how chronically denervated the muscle is.

## Figures and Tables

**Figure 1 diagnostics-13-00219-f001:**
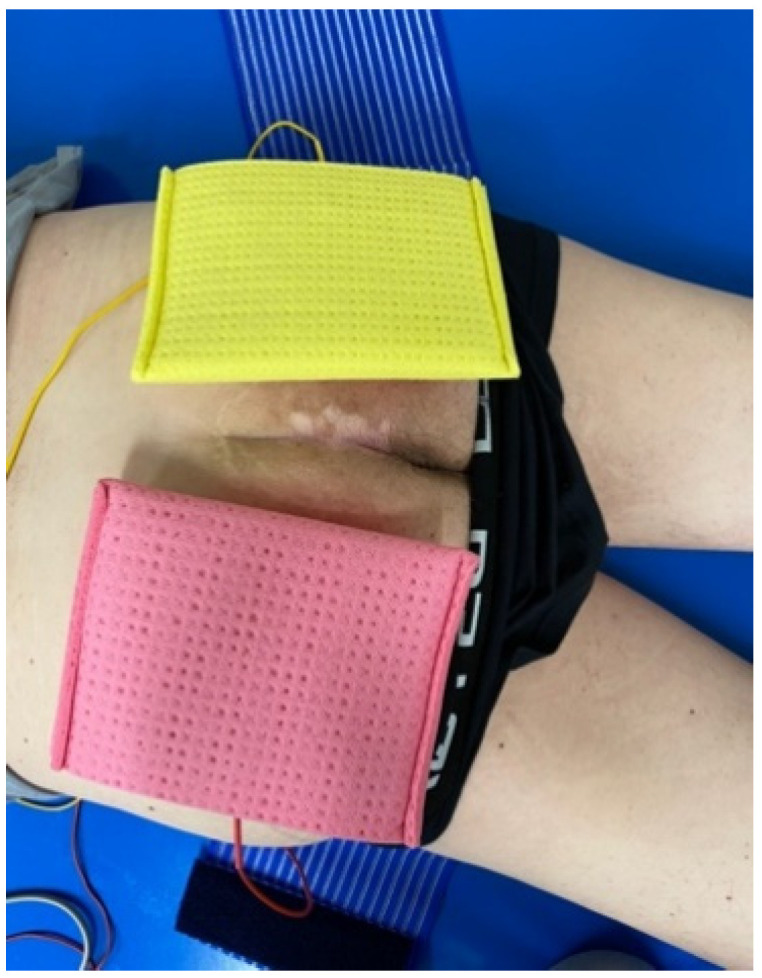
Stimulation set up with sponge pockets including electrodes.

**Figure 2 diagnostics-13-00219-f002:**
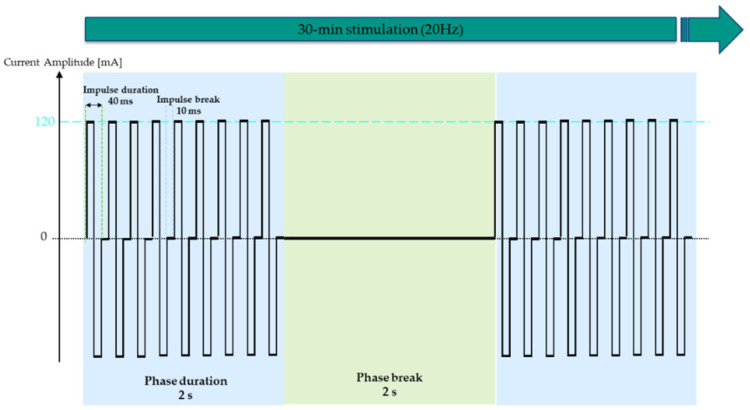
Electrical stimulation parameters of the 30-min training session, as administered after 3-min warm-up. Symmetrical biphasic rectangular current with 2 s bursts consisting of 40 ms impulses alternated with 10 ms impulse breaks. Stimulation intensity was kept at 120 mA throughout the study for all participants.

**Figure 3 diagnostics-13-00219-f003:**
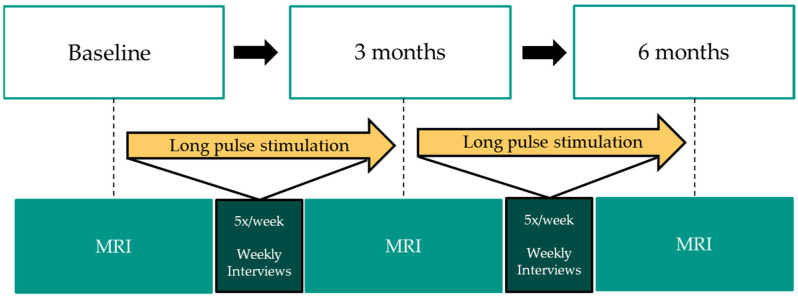
Study participation workflow.

**Figure 4 diagnostics-13-00219-f004:**
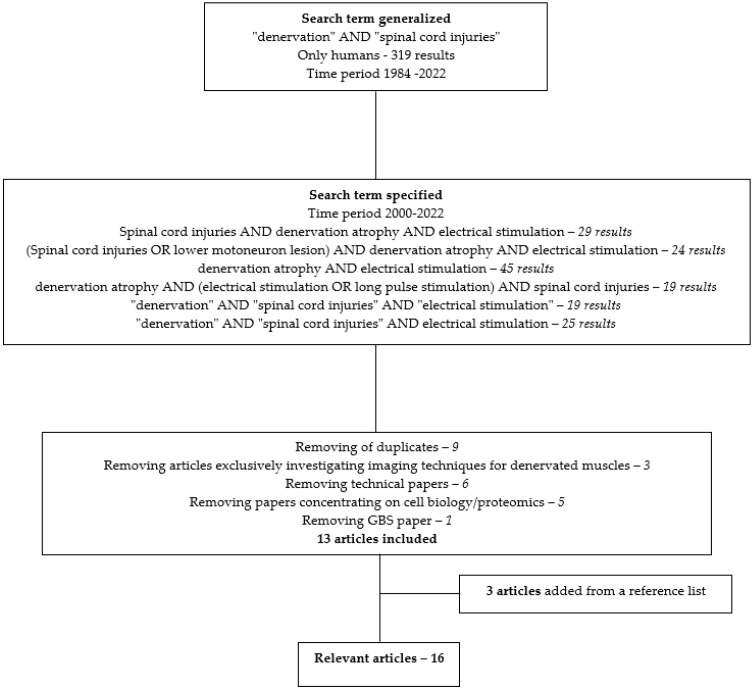
Flow chart of narrative literature search.

**Figure 5 diagnostics-13-00219-f005:**
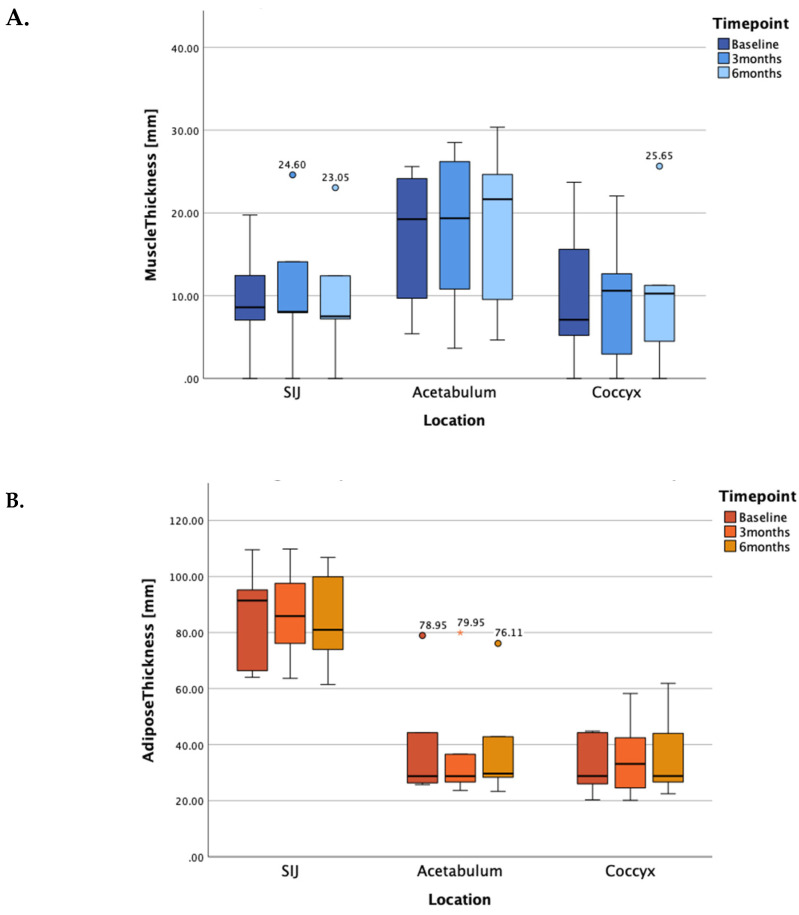
Median and interquartile ranges of muscle (panel **A**) and adipose (panel **B**) tissue thicknesses at the levels of the SIJ, the acetabulum and at the coccyx at baseline, 3 months and 6 months after stimulation onset, respectively. SIJ: sacroiliac joint.

**Table 1 diagnostics-13-00219-t001:** Participant demographics.

Participant	Age	NLI	AIS Score	Time Since Injury (Years)	Previous Pressure Ulcers (Number, Location, Treatment)	Stimulation Period & Number of Sessions
1	43	T9	A	23	Recuring, sacrum + tuber ischiadicum, conservative	21 weeks, 92 sessions
2	49	L1	C	2	3, tuber ischiadicum, operative	26 weeks, 129 sessions
3	33	T11	A	32	2, tuber ischiadicum, conservative	26 weeks, 117 sessions
4	65	C4	A	27	2, tuber ischiadicum, conservative	27 weeks, 129 sessions
5	62	T12	A	20	No data	27 weeks, 121 sessions

AIS: American Spinal Injury Association Impairment Scale; NLI: neurological level of injury; weeks: weeks. All pressure ulcers reported in the patient history were rated at least grade 3.

## Data Availability

Data will be made available upon request to the corresponding author.

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
