# Peer review of "Electrical Stimulation for Preventing Skin Injuries in Denervated Gluteal Muscles—Promising Perspectives from a Case Series and Narrative Review"

_diagnostics, 2023, doi:10.3390/diagnostics13020219_

Round 1

Reviewer 1 Report

This paper addresses an important topic: Muscle atrophy and degeneration due to lower motor neuron lesions related to spinal cord injury, and as a consequence, increased risk for decubitus ulcers related to the loss of cushioning effect. Specifically, the authors assess gluteal muscle and adipose tissue thickness using magnetic resonance images following long-pulse duration, home-based electrical stimulation of the muscle over six months. The topic is relevant; however, the paper needs clarifications and revisions to meet the goal with the Diagnostics readers in its current draft, as listed in the comments and suggestions for the authors below:

Major:

Materials and Methods:

·         Please add inclusion/exclusion criteria to the manuscript.

·         Add a detailed explanation of how the lower motor neuron lesion was diagnosed.

·         Elaborate on subjects’ pressure sore history.

·         On the ClinicalTrials.gov webpage, NCT0280039 properly states that one of the purposes and secondary outcomes of this study is to evaluate the distribution of seating pressure. This outcome is important for the assessment of pressure ulcer risk. However, the authors did not mention this outcome in the methods or results section of the manuscript. Please clarify.

Results:

·         Please discuss the selected articles’ findings instead of summarizing the results in the appendix.

·         Elaborate on whether the subjects were compliant with the stimulation protocol, i.e., 5 times per week for 6 months, and how you controlled it. For example, did all participants get the same amount of stimulation?

·         The study has five subjects; one is two years post-injury and the other 32 years post-injury. Therefore, it may be more informative to the reader to show each subject’s measurements separately on the same plot besides median and interquartile ranges. Then, the results may be discussed whether the time since injury affects the results.

Minor:

·         The first time the term is used, put the acronym in parentheses after the full term. (see lines 51 and 55)

·         Line 37: “All of these factors taken together and …….” This sentence is not neat in terms of language. Please consider rephrasing.

·         Table 1- Please replace “ASIA Score” with “AIS score,” and the table footnote as “AIS: American Spinal Injury Association Impairment Scale”

·         The sentences in Lines 112 and 121 are repeated sentences.

·         A few errata/typo (corrections):

Line 13: …. 30 years has been making ….. (…. 30 years has been made …..)

Line 28: … spinal cord injury (SCI) have … (… spinal cord injury (SCI) has …)

Appendix, page 12: Traumatic conus cauna …. (Traumatic conus cauda ….)

Author Response

Dear reviewer,

Thanks for your valuable comments to improve the quality of our article. We carefully revised our manuscript and changed it according to your suggestions. Please find our replies/comments below:

Comment:

Materials and Methods:

Please add inclusion/exclusion criteria to the manuscript.

Add a detailed explanation of how the lower motor neuron lesion was diagnosed.

Elaborate on subjects’ pressure sore history.

Reply:

Thank you for these comments.

  • We have added the in- and exclusion criteria in the methods section
  • We have added a description of what is identified as a LMN lesion to our manuscript.
  • After other reviewers’ comments, we have shifted the participant demographics table to the results section, where we have complemented the table with information about pressure ulcer history, grade and treatment.

Comment:

On the ClinicalTrials.gov webpage, NCT0280039 properly states that one of the purposes and secondary outcomes of this study is to evaluate the distribution of seating pressure. This outcome is important for the assessment of pressure ulcer risk. However, the authors did not mention this outcome in the methods or results section of the manuscript. Please clarify.

Reply:

Thank you for addressing this point. We agree that this is an important outcome to evaluate pressure ulcer risk. As this is a study of preliminary results and as the data on the pressure distribution were very heterogeneous, we have decided to wait until more participants will be included to analyze the full data set.

Comment:

Results:

Please discuss the selected articles’ findings instead of summarizing the results in the appendix.

Elaborate on whether the subjects were compliant with the stimulation protocol, i.e., 5 times per week for 6 months, and how you controlled it. For example, did all participants get the same amount of stimulation?

The study has five subjects; one is two years post-injury and the other 32 years post-injury. Therefore, it may be more informative to the reader to show each subject’s measurements separately on the same plot besides median and interquartile ranges. Then, the results may be discussed whether the time since injury affects the results.

Reply:

  • Thank you for this advice. However, we think that an overview of the articles' findings in the appendix table served the purpose of including it in the paper. We also integrated some of the literature in the introduction and compared our findings to previous work retrieved for the narrative review.
  • Thanks for commenting on the dose control. We have given an overview of accomplished stimulation sessions in the results section (table 1).
  • Thanks for this advice. We have tried displaying the data in different manners. However, in our opinion, the way we have done it in the first manuscript version gives the best overview. That is why we have decided to maintain the present version.

Minor

Comment:

The first time the term is used, put the acronym in parentheses after the full term. (see lines 51 and 55)

Reply: We made the changes according to your comment.

Comment:

Line 37: “All of these factors taken together and …….” This sentence is not neat in terms of language. Please consider rephrasing.

Reply:

We have rephrased this sentence to: Considering the multitude of risk factors present in people with SCI, and the fact that a majority if wheelchair-bound, a LMN lesion in the gluteal region puts them at a major risk of developing tissue damage.

Comment:

Table 1- Please replace “ASIA Score” with “AIS score,” and the table footnote as “AIS: American Spinal Injury Association Impairment Scale”

Reply:

Thank you for correcting this, we have made the changes accordingly.

Comment:

The sentences in Lines 112 and 121 are repeated sentences.

Reply:

We have deleted the sentence in line 112 so that the first paragraph is dedicated to positioning of patient and equipment and the next section includes the technical parameters of the stimulation.

Comments:

A few errata/typo (corrections):

Line 13: …. 30 years has been making ….. (…. 30 years has been made …..)

Line 28: … spinal cord injury (SCI) have … (… spinal cord injury (SCI) has …)

Appendix, page 12: Traumatic conus cauna …. (Traumatic conus cauda ….)

Reply:

Thank you, we have removed all typos:

  • Line 13: previous research over the last 30 years has made advancements in the development of parameters
  • Line 28: have -> has
  • Page 12: conus cauna -> conus cauda

Reviewer 2 Report

Dear Authors,

Spinal cord injury is a highly prevalent and complex disease, which is associated with a detrimental burden of comorbidities and factors contributing to worsening health outcomes. Among the complications that the patients might experiment, pressure ulcers contribute to high hospitalization rates, thus worsening the patients’ health outcomes.

In this context, providing an effective preventive strategy could have intriguing implications for the clinical management of the patients.

However, I have some concerns that should be assessed in order to make this paper suitable for publication.

Major reviews

INTRODUCTION. The whole section should be implemented by citing more supporting evidence. Please, find some examples here enclosed.

-       Invernizzi M, de Sire A, Fusco N. Rethinking the clinical management of volumetric muscle loss in patients with spinal cord injury: Synergy among nutritional supplementation, pharmacotherapy, and rehabilitation. Curr Opin Pharmacol. 2021 Apr;57:132-139. doi: 10.1016/j.coph.2021.02.003. Epub 2021 Mar 12. PMID: 33721616.

-       Greising SM, Dearth CL, Corona BT. Regenerative and Rehabilitative Medicine: A Necessary Synergy for Functional Recovery from Volumetric Muscle Loss Injury. Cells Tissues Organs. 2016;202(3-4):237-249. doi: 10.1159/000444673. Epub 2016 Nov 9. PMID: 27825146; PMCID: PMC5553044.

MATERIALS AND METHODS. Please, provide information about the study duration.

MATERIALS AND METHODS. Apparently, inclusion and exclusion criteria are missing. Please, implement the section consequently.

MATERIALS AND METHODS. Study participants should be better characterized by providing information about their anamnesis, clinical history, comorbidities, medications, a preliminary nutritional assessment, weight and BMI measurement, functional level assessed with relevant scales, and any ongoing or previously occurred pressure ulcer (especially in gluteal region).

Moreover, was ASIA assessment derived from anamnestic data or performed upon enrolment? If so, information about the assessor(s) should be provided (for instance, was the operator a certified assessor? Years of experience?)

MATERIALS AND METHODS. Please, provide more information about the procedure itself. Did the participant underwrite informed consent? Were the participants recruited as outpatients or on territory?

RESULTS. The section should be implemented by reporting data about the participants (i.e. Table 1 should be moved under the results heading).

DISCUSSION. The section should be implemented taking into account any effort performed in excluding the potential bias and sources of confoundment which could have affected the prospected results. For instance, literature shows that weight, BMI and nutritional implication are important factors to take into account when assessing pressure ulcers and muscular thickness.

-       Šín P, Hokynková A, Marie N, Andrea P, Krč R, Podroužek J. Machine Learning-Based Pressure Ulcer Prediction in Modular Critical Care Data. Diagnostics (Basel). 2022 Mar 30;12(4):850. doi: 10.3390/diagnostics12040850. PMID: 35453898.

-       de Sire A, Ferrillo M, Lippi L, Agostini F, de Sire R, Ferrara PE, Raguso G, Riso S, Roccuzzo A, Ronconi G, Invernizzi M, Migliario M. Sarcopenic Dysphagia, Malnutrition, and Oral Frailty in Elderly: A Comprehensive Review. Nutrients. 2022 Feb 25;14(5):982. doi: 10.3390/nu14050982. PMID: 35267957; PMCID: PMC8912303.

In addition, further stratification in risk of development of pressure ulcers might be assessed basing on the level of SCI.

Moreover, supervision might positively affect the correctness of the ES protocol: I think that you should provide further details about this aspect.

DISCUSSION. MRI periodical scans for tissue composition are the optimal procedure, but some limitations arise in terms of time elapsed and costs; moreover, the participants might experiment contraindications to MRI (i.e. the presence of ferromagnetic foreign bodies, claustrophobia) or difficulties in transportations for the periodical follow-up and could be lost during follow-up for the reasons above. Do you think that US assessment might provide similar data integrated into clinical settings?

Please discuss.

Minor reviews

INTRODUCTION. Page 1, line 28. Please, change the word “have” with “has”.

INTRODUCTION. Page 2, line 51. Please, note that explanation for abbreviations (i.e. ES) should be provided on the first appearance of the abbreviation.

INTRODUCTION. Page 2, lines 51-54. Please, note that any information about the study design and the procedures performed should be reported in the “Methods” section.

INTRODUCTION. Page 3, lines 67-69; lines 76-78; lines 82-86. These periods’ syntax should be reviewed. Please, rephrase them accordingly.

INTRODUCTION. Page 3, line 74. Please, change “As has been shown […]” with “As it has been shown […]”.

MATERIALS AND METHODS. Page 4, line 113. Please, change “is” with “was”.

TABLE 1. Please, move the line number (109) on right.

Author Response

Dear reviewer,

Thanks for your valuable comments to improve the quality of our article. We carefully revised our manuscript and changed it according to your suggestions.

Comment: INTRODUCTION. The whole section should be implemented by citing more supporting evidence. Please, find some examples here enclosed.

-       Invernizzi M, de Sire A, Fusco N. Rethinking the clinical management of volumetric muscle loss in patients with spinal cord injury: Synergy among nutritional supplementation, pharmacotherapy, and rehabilitation. Curr Opin Pharmacol. 2021 Apr;57:132-139. doi: 10.1016/j.coph.2021.02.003. Epub 2021 Mar 12. PMID: 33721616.

-       Greising SM, Dearth CL, Corona BT. Regenerative and Rehabilitative Medicine: A Necessary Synergy for Functional Recovery from Volumetric Muscle Loss Injury. Cells Tissues Organs. 2016;202(3-4):237-249. doi: 10.1159/000444673. Epub 2016 Nov 9. PMID: 27825146; PMCID: PMC5553044.

Reply: We would like to thank you for addressing this important factor. We have implemented your suggestion to our work by adding a sentence and reference to the introduction.

Comment: MATERIALS AND METHODS. Please, provide information about the study duration.

Reply: We have added the start of recruitment of the study recruitment phase to our manuscript (March 2020). As it is an ongoing study, we cannot predict the conclusion date.

Comment: MATERIALS AND METHODS. Apparently, inclusion and exclusion criteria are missing. Please, implement the section consequently.

Reply: Thanks, this is essential information indeed. In- and exclusion criteria have been added in the methods section.

Comment: MATERIALS AND METHODS. Study participants should be better characterized by providing information about their anamnesis, clinical history, comorbidities, medications, a preliminary nutritional assessment, weight and BMI measurement, functional level assessed with relevant scales, and any ongoing or previously occurred pressure ulcer (especially in gluteal region).

Moreover, was ASIA assessment derived from anamnestic data or performed upon enrolment? If so, information about the assessor(s) should be provided (for instance, was the operator a certified assessor? Years of experience?)

Reply: Thank you for your comment on this. Regarding the patient history, we have added anamnestic information that is relevant to the investigated topic. Hence, please find information regarding preceding decubitus and the grade of severity in table 1 of patient demographics (which has been shifter to results).

We would like to add that the Swiss Paraplegic Centre is a specialized hospital for people with spinal cord injury where all patients are screened on a regular basis according to the international standards for neurological classification by an EMSCI-trained and experienced neurologist. Additionally, the clinic is also providing data to the EMSCI-database.

Comment: MATERIALS AND METHODS. Please, provide more information about the procedure itself. Did the participant underwrite informed consent? Were the participants recruited as outpatients or on territory?

Reply: We completed and implemented more information about recruitment procedure and informed consent.

Comment: RESULTS. The section should be implemented by reporting data about the participants (i.e. Table 1 should be moved under the results heading).

Reply: We added more information about the patient characteristics as you proposed in the previous comment. In addition, we have shifted table 1 into the result section.

Comment: DISCUSSION

Moreover, supervision might positively affect the correctness of the ES protocol: I think that you should provide further details about this aspect.

Reply: We have completed this in the methodological description above figure 3 – workflow.

Comment: DISCUSSION. The section should be implemented taking into account any effort performed in excluding the potential bias and sources of confoundment which could have affected the prospected results. For instance, literature shows that weight, BMI and nutritional implication are important factors to take into account when assessing pressure ulcers and muscular thickness.

-       Šín P, Hokynková A, Marie N, Andrea P, Krč R, Podroužek J. Machine Learning-Based Pressure Ulcer Prediction in Modular Critical Care Data. Diagnostics (Basel). 2022 Mar 30;12(4):850. doi: 10.3390/diagnostics12040850. PMID: 35453898.

-       de Sire A, Ferrillo M, Lippi L, Agostini F, de Sire R, Ferrara PE, Raguso G, Riso S, Roccuzzo A, Ronconi G, Invernizzi M, Migliario M. Sarcopenic Dysphagia, Malnutrition, and Oral Frailty in Elderly: A Comprehensive Review. Nutrients. 2022 Feb 25;14(5):982. doi: 10.3390/nu14050982. PMID: 35267957; PMCID: PMC8912303.

In addition, further stratification in risk of development of pressure ulcers might be assessed basing on the level of SCI.

Reply: We added a section to the introduction highlighting potential risk factors for developing pressure injuries. (Shiferaw, W. S., Akalu, T. Y., Mulugeta, H. & Aynalem, Y. A. The global burden of pressure ulcers among patients with spinal cord injury: a systematic review and meta-analysis. Bmc Musculoskelet Di 21, 334 (2020). These defined risk factors are in accordance with the worldwide non-SCI population developing a pressure injury (Šín P, Hokynková A, Marie N, Andrea P, Krč R, Podroužek J. Machine Learning-Based Pressure Ulcer Prediction in Modular Critical Care Data. Diagnostics (Basel). 2022 Mar 30;12(4):850. doi: 10.3390/diagnostics12040850. PMID: 35453898.)

Comment:

DISCUSSION. MRI periodical scans for tissue composition are the optimal procedure, but some limitations arise in terms of time elapsed and costs; moreover, the participants might experiment contraindications to MRI (i.e. the presence of ferromagnetic foreign bodies, claustrophobia) or difficulties in transportations for the periodical follow-up and could be lost during follow-up for the reasons above. Do you think that US assessment might provide similar data integrated into clinical settings? Please discuss.

Reply: Thank you, we absolutely agree. For simplicity and financial reasons, it is true that ultrasound is a useful tool for clinical assessment of muscle and adipose tissue thickness that we also use in a standardized way for muscle morphological assessment in the patient population seen at our department (SCI with upper motor neuron lesions, orthopedics,…). However, MRI offers a much higher sensitivity in depicting early changes in tissue composition of people with chronic SCI and widely degenerated muscles. This is the reason why we opted for this method and why this has been established and recognized as an assessment tool by specialists in the field in the past ten years.

Comment: Minor reviews

Replies:

INTRODUCTION. Page 1, line 28. Please, change the word “have” with “has”.

- has been changed

INTRODUCTION. Page 2, line 51. Please, note that explanation for abbreviations (i.e. ES) should be provided on the first appearance of the abbreviation.

- The explanation for abbreviation precedes the abbreviation at first appearance

INTRODUCTION. Page 2, lines 51-54. Please, note that any information about the study design and the procedures performed should be reported in the “Methods” section.

- All procedures about the actual study that we are conducting are stated and described in the “methods” section. We would like to keep information about our literature search in the introduction as it prepares the reader for the subject.

INTRODUCTION. Page 3, lines 67-69; lines 76-78; lines 82-86. These periods’ syntax should be reviewed. Please, rephrase them accordingly.

- all three sentences were reviewed and adapted for better understanding.

INTRODUCTION. Page 3, line 74. Please, change “As has been shown […]” with “As it has been shown […]”.

- we have added the word “it”

MATERIALS AND METHODS. Page 4, line 113. Please, change “is” with “was”.

- We have made the replacement

TABLE 1. Please, move the line number (109) on right.

- number has been moved in table 1, that has been shifted to the results section and completed with missing relevant information about previous pressure ulcers

hopefully it will appear, we think there is a template issue that we cannot influence

Reviewer 3 Report

10 the verb is missing

11-12 is not clear. after several re-readings it becomes evident that affected muscles can lead to lesions and skin disorders.

15 resuscitate? to recover..

the methods of the abstract are missing, it seems that there are 3 paraphrases of the goal. Is the manuscript a case series with a review of the literature? put at least population, intervention and outcome. Regarding the electrical stimulation, there are several and different in detail physical agent modalities, it is not enough to talk about long pulse. I believe this is NMES

In the results put data

27-30 missing references

Removal by convention from the introduction figure and results of a review. It is not necessary. describe the background in a discursive way, also because you want to give preliminary results ..

In methods, describe the methodology not the results.

Design with the features of this rapid review and how it was conducted, remove the case series results, but instead describe the population you are going to select. , age, age of SCI, cut-off ASIA? describe the methods of intervention and outcome measures. A Power Analysis of the future trial?

In the results show the table with the characteristics of the included studies and therefore how you addressed the intervention to your 5 patients.

There are perspectives in the literature, but the limitations of your study are missing

Author Response

Dear reviewer,

Thanks for your comments to improve the quality of our article. We carefully revised our manuscript and changed it according to your suggestions. Please find our replies to the individual comments below:

Comment: 10 the verb is missing

Reply: We are sorry, but we cannot find the mistake of a missing verb in this sentence:
Spinal cord injury (SCI) where the lower motor neuron is compromised leads to atrophy and degenerative changes in the respective muscle
If we are not talking about the same sentence, please let us know.

Comment: 11-12 is not clear. after several re-readings it becomes evident that affected muscles can lead to lesions and skin disorders.

Reply: Indeed, the thicker the muscle tissue, the more of a “cushioning” it offers between bone and skin. To clarify, the state of the muscle itself does not lead to skin disorder but rather the mechanical consequences resulting from its atrophy. We hope that this information can clarify the sentence and if not, please let us know what exactly needs more elucidation.

Comment: 15 resuscitate? to recover..

Reply: Thank you for your suggestion. We have replaced the word “resuscitate” by “restore” as the recovery suggests a healing process. However, we solely affect the muscle structure without restoring function.

Comment: the methods of the abstract are missing, it seems that there are 3 paraphrases of the goal. Is the manuscript a case series with a review of the literature? put at least population, intervention and outcome. Regarding the electrical stimulation, there are several and different in detail physical agent modalities, it is not enough to talk about long pulse. I believe this is NMES

Reply: Yes, the manuscript is composed of a small literature review to give an overview of the treatment method applying long pulse stimulation in people with SCI and damage to the lower motoneuron of the gluteal muscles. As we have mentioned the methods used (line 18) and the population (line 18) in the abstract, we have completed it by stating the number of participants included for the present manuscript. We do not think that stating more detailed information about the reviewed literature is necessary to be stated in the abstract. We will keep the terminology long pulse stimulation. It describes exactly the method and is used in the current literature to clarify the stimulation of denervated muscles (lower motoneuron lesion). Hereby the pulse duration ranges from 10 – 1000 ms. The term NMES describes the neuromuscular electrical stimulation, a stimulation conducted via nerve with a pulse duration of 250-600 ms.

Comment: In the results put data

Reply: Thank you for your comment. Would you suggest adding data to the graph to help clarify the changes? We would like to point out that all relevant data has been stated and described in the text above the figure. We would like to fulfill your suggestion, but it remains unclear to us what kind of data you are missing in our result section? Please clarify.

Comment: 27-30 missing references

Reply: Thank you for mentioning. We added an additional reference.

Comment: Removal by convention from the introduction figure and results of a review. It is not necessary. describe the background in a discursive way, also because you want to give preliminary results ..

Reply: We are very sorry, but we do not quite understand what you would like to express with this comment. Do you want to say that the literature review is not necessary to be declared and displayed as such in the present work?

Comment: In methods, describe the methodology not the results.

Reply: Thank you for this remark. We have shifted table 1 to the results section.

Comment: Design with the features of this rapid review and how it was conducted, remove the case series results, but instead describe the population you are going to select. , age, age of SCI, cut-off ASIA? describe the methods of intervention and outcome measures. A Power Analysis of the future trial?

Reply: We would like to point out that the review is not the primary focus of this work. It should give the reader a short and concise overview of the most important research in the field that has been conducted over the last 30 years regarding SCI, denervation and changing muscle mass through application of home-based long pulse ES. Hence, we would like to describe the methodology and in-and exclusion criteria applying to our study. We have complemented the methods section in this aspect. After completion of the pilot study a power analysis for a larger clinical trial will be performed.

Comment: In the results show the table with the characteristics of the included studies and therefore how you addressed the intervention to your 5 patients.

Reply: We have only conducted this small narrative review as background information and to give an overview of what has been done in the past. We also based our parameter selection for the intervention on previous published recommendations. The table with demographics of the included patients has been moved to results. The intervention and way of application has been extensively described in the methods section.

Comment: There are perspectives in the literature, but the limitations of your study are missing

Reply: Thank you, the limitations have not been included under a separate heading. Please find them in the paragraph from line 302-314.

Round 2

Reviewer 1 Report

None

Reviewer 2 Report

None

Reviewer 3 Report

Dear authors, I suggest more methodological clarity and to add: “A Case Series and Literature Review” in the title

Abstract

I was referring to the abstract. In the methods make it clear that your work is a literature review with a case series. In the methods defined the intervention and the outcome. Where have they been stimulated for how long? What measurements were used? In the results you need to put some data (+2, -7? ES? Friedman test?), have they improved? How, but above all how much?

Introduction

Remove figure 1, from a methodological point of view in the introduction you cannot put the results of a review of the literature, if you want to keep the figure and methodological approach, move the search strategy into the methods, move the figure into the results. Otherwise, only provide a narrative character, but I do not recommend it also because in the objective at the end of the introduction you aim to do a review and present these case series ...

Methods

138 has anyone already used these parameters?

Divide the methods as Participants with eligibility, Intervention with the parameters and positions of the devices and finally the outcomes ... (among other things, has anyone else used MRI as an assessment? participant patterns, intervention, outcome), the evaluation scales shown in the results are missing.

Statistical analysis requires a separate paragraph by convention. It’s necessary non-parametric test

Results

It is true that the median and the interquartile range are not influenced by the extreme values, but they are still 5 patients .. it would be interesting to evaluate the individual participants and for example consider the role of ES in patient 2 compared to people suffering from SCI from more than 20 years.T

here are perspectives in the literature, but the limitations of your study are missing
